# HIV-1 vaccination by needle-free oral injection induces strong mucosal immunity and protects against SHIV challenge

Andrew T. Jones[1,2], Xiaoying Shen[3], Korey L. Walter [4], Celia C. LaBranche[5], Linda S. Wyatt[6], Georgia D. Tomaras[3], David C. Montefiori[5], Bernard Moss[6], Dan H. Barouch[7], John D. Clements[8], Pamela A. Kozlowski[4], Raghavan Varadarajan[9] & Rama Rao Amara [1,2]

The oral mucosa is an attractive site for mucosal vaccination, however the thick squamous epithelium limits antigen uptake. Here we utilize a modified needle-free injector to deliver immunizations to the sublingual and buccal (SL/B) tissue of rhesus macaques. Needle-free SL/B vaccination with modified vaccinia Ankara (MVA) and a recombinant trimeric gp120 protein generates strong vaccine-specific IgG responses in serum as well as vaginal, rectal and salivary secretions. Vaccine-induced IgG responses show a remarkable breadth against gp70-V1V2 sequences from multiple clades of HIV-1. In contrast, topical SL/B immunizations generates minimal IgG responses. Following six intrarectal pathogenic SHIV-SF162P3 challenges, needle-free but not topical immunization results in a significant delay of acquisition of infection. Delay of infection correlates with non-neutralizing antibody effector function, Env-specific CD4$^+$ T-cell responses, and gp120 V2 loop specific antibodies. These results demonstrate needle-free MVA/gp120 oral vaccination as a practical and effective route to induce protective immunity against HIV-1.

[1] Emory Vaccine Center, Yerkes National Primate Research Center, Emory University, Atlanta, GA 30329, USA. [2] Department of Microbiology and Immunology, Emory School of Medicine, Emory University, Atlanta, Georgia 30329, USA. [3] Duke Human Vaccine Institute, Duke University Medical Center, Durham, NC 27710, USA. [4] Department of Microbiology, Immunology and Parasitology, Louisiana State University Health Sciences Center, New Orleans, LA 70112, USA. [5] Department of Surgery, Duke University Medical Center, Durham, NC 27710, USA. [6] Laboratory of Viral Diseases, National Institute of Allergy and Infectious Diseases, National Institutes of Health, Bethesda, MD 20892, USA. [7] Center for Virology and Vaccine Research, Beth Israel Deaconess Medical Center, Harvard Medical School, Boston, MA 02115, USA. [8] Department of Microbiology and Immunology, Tulane University School of Medicine, New Orleans, LA 8638, USA. [9] Molecular Biophysics Unit, Indian Institute of Science, Bangalore 560012, India. Correspondence and requests for materials should be addressed to R.R.A. (email: ramara@emory.edu)

Human immunodeficiency virus-1 (HIV-1) is most commonly transmitted across genital and rectal mucosal surfaces via sexual contact[1]. Within the first days and weeks of infection, HIV-1 is localized to the mucosal tissue, replicating in resident target cells, before systemic dissemination and seroconversion[2]. In addition, irrespective of the route of infection, HIV causes a rapid and profound depletion of CD4 T cells in the gut[3]. Because of this, genital and gut mucosal immunity against HIV-1 is crucial in combating the virus in this early state. Mucosal vaccination, in which immunizations are delivered directly to the mucosal tissues, are the most effective method of generating mucosal immunity[4]. While mucosal vaccines for HIV-1 have been investigated in non-human primate models, few human clinical trials have evaluated mucosal HIV-1 vaccination, and mucosal responses have rarely been characterized in previous clinical trials[4,5].

Oral vaccines are attractive as they can induce strong immunity in the gut, are relatively non-invasive, and can be administered on a large scale[4]. Oral vaccines generally are ingested and thus must survive the hostile acidic environment of the stomach to be sampled by the gut-associated lymphoid tissue (GALT) mainly in the distal regions of the small intestine. An alternative strategy of oral vaccination is to directly target the tissues within the oral mucosa for antigen delivery. Vaccination of the oral mucosa, primarily to the buccal (inner cheek) and sublingual (below the tongue) tissue, has been proposed to be a practical, safe, and non-invasive route of mucosal vaccination[6,7]. The sublingual and buccal (SL/B) tissues contain numerous subsets of antigen-presenting cells, however these populations have not been fully characterized in humans and non-human primates[8]. Most sublingual and buccal vaccinations are performed by the topical application of vaccines to oral tissues, allowing for natural absorption across the oral epithelium. The oral mucosa, unlike the simple columnar epithelium in the small intestine, consists of multilayered squamous epithelium, which can limit the natural uptake of vaccine antigens. Thus, oral vaccination approaches that enhance vaccine uptake may significantly increase vaccine responses.

To aid in antigen uptake, needle-free injectors can be used to deliver vaccines across the skin or oral epithelium into the underlying tissue, while retaining the non-invasive features of oral vaccination[9,10]. Needle-free injectors have been investigated as a tool to deliver drugs and vaccinations, primarily through the skin, and are an attractive alternative to needle-stick based injections which carry disadvantages such as the need for trained healthcare workers to administer injects, the risks associated with needle-sticks and re-using needles, as well as the common fear of needles resulting in reduced patient compliance[10–12].

Here we evaluate the SL/B tissue as a route of needle-free oral HIV-1 vaccination in rhesus macaques. Vaccine components are delivered orally to the SL/B tissues via either needle-free injection (Needle-free SL/B) or topical application (Topical SL/B) and compared to the conventional intradermal/subcutaneous route (ID/SC) commonly used for needle-based immunizations. Vaccinations consist of two priming immunizations with modified vaccinia Ankara (MVA) engineered to express HIV-1 antigens (MVA-HIV) followed by boosting twice with a recombinant trimeric gp120 immunogen (cycP-gp120), along with the *Escherichia coli* derived mucosal adjuvant double mutated heat-labile enterotoxin (dmLT), which has been shown to promote mucosal immune responses[13]. MVA-HIV has been extensively characterized in non-human primate studies and is currently being tested in human clinical trials as an HIV-1 vaccine candidate[14–16], and cycP-gp120 has previously been shown to elicit tier-2 neutralizing antibodies in guinea pigs as well as promote highly-cross reactive V1V2-directed antibodies, a major correlate

of protection in the RV144 trial, in rabbits and rhesus macaques[17–19]. To test the vaccine efficacy of MVA-HIV/cycP-gp120, animals are challenged intra-rectally 19 weeks after the last immunization with a heterologous, pathogenic, difficult to neutralize SHIV-SF162P3. Needle-free SL/B and ID/SC immunization both results in a significant delay in acquisition of infection compared to unvaccinated controls, with non-neutralizing antibody effector functionality and Env-specific CD4 T-cell responses being the predominant correlates of protection. These results demonstrate that oral needle-free delivery of MVA-HIV/cycP-gp120 vaccination induces a strong antibody response in mucosal and systemic compartments with protective potential against HIV-1. They also describe a novel method of needle-free vaccination that is practical and induces a strong antibody response in three major mucosal sites and serum.

## Results

**Dendritic cells in the sublingual and buccal tissues.** While antigen-presenting cells in the sublingual and buccal tissue (SL/B) have been described for mice and humans[8], they have not been characterized well in rhesus macaques. Langerhans cells (LCs) are a major subset of the migratory dendritic cells in the epidermis and have multiple roles in inducing immune responses, from maintaining immune homeostasis to activation and presentation of antigens upon inflammation[20,21]. To determine the presence of LCs in the sublingual and buccal tissue, we stained paraffin-embedded sections of rhesus macaque sublingual and buccal tissue with hematoxylin and eosin (H&E) to characterize the overall architecture of the oral epithelium and underlying tissue (Fig. 1a), and performed immunohistochemistry with an anti-langerin antibody (Fig. 1b). LCs could be detected in both sublingual and buccal tissue, primarily residing in the mucosal epithelium, confirming the presence of these cells within the oral tissue of rhesus macaques. To further characterize dendritic cell subsets in the sublingual and buccal tissue, we prepared single cell suspensions from biopsied samples and analyzed via flow cytometry. Antigen-presenting cells were first gated as HLA-DR$^+$CD45$^+$CD3$^-$CD20$^-$live cells (Supplementary Figure 1). In both the sublingual and buccal tissue, we detected a population of CD14$^+$DC-SIGN$^+$ dendritic or macrophage-like cell (dermal DCs)[20,22] (Fig. 1c). Additionally, we detected conventional BDCA1$^+$ myeloid dendritic cells (cDCs) (CD14$^-$CD16$^-$CD123$^-$BDCA1$^+$), but only low levels of plasmacytoid dendritic cells (pDCs) (CD14$^-$CD16$^-$CD123$^+$BDCA1$^-$) which may only be present during inflammation[8]. Conversely, we detected much greater levels of pDCs present in the submandibular and submental lymph nodes that drain oral tissue. These lymph nodes also contained both dermal DCs and cDCs, though at a lower proportion to the buccal and sublingual tissue (Fig. 1c). Interestingly, we found the buccal tissue contained a significantly higher proportion of both cDCs (BDCA-1$^+$CD123$^-$) and dermal DCs (CD14$^+$DC-SIGN$^+$), compared to the sublingual tissue. These data show there is a diverse population of antigen-presenting cells in the SL/B tissue and the local draining lymph nodes. Our efforts to stain for LCs using flow cytometry were not successful likely due to poor binding of the antibody in this setting.

The vitamin A metabolite, retinoic acid, is important for generating gut-homing adaptive immune responses[23,24]. Dendritic cells process vitamin A to retinoic acid via retinal dehydrogenase (RALDH) enzymes[25,26], the activity of which can be detected using the ALDEFLUOR$^{TM}$ kit (STEMCELL). We characterized the expression of RALDH enzymes by the APCs in the sublingual and buccal tissues as well oral mucosa draining lymph nodes using the ALDEFLUOR$^{TM}$ reagent and flow cytometry. We found the RALDH activity in both BDCA1$^+$CD123$^-$ cDCs and CD14$^+$DC-SIGN$^+$ dermal DCs present in

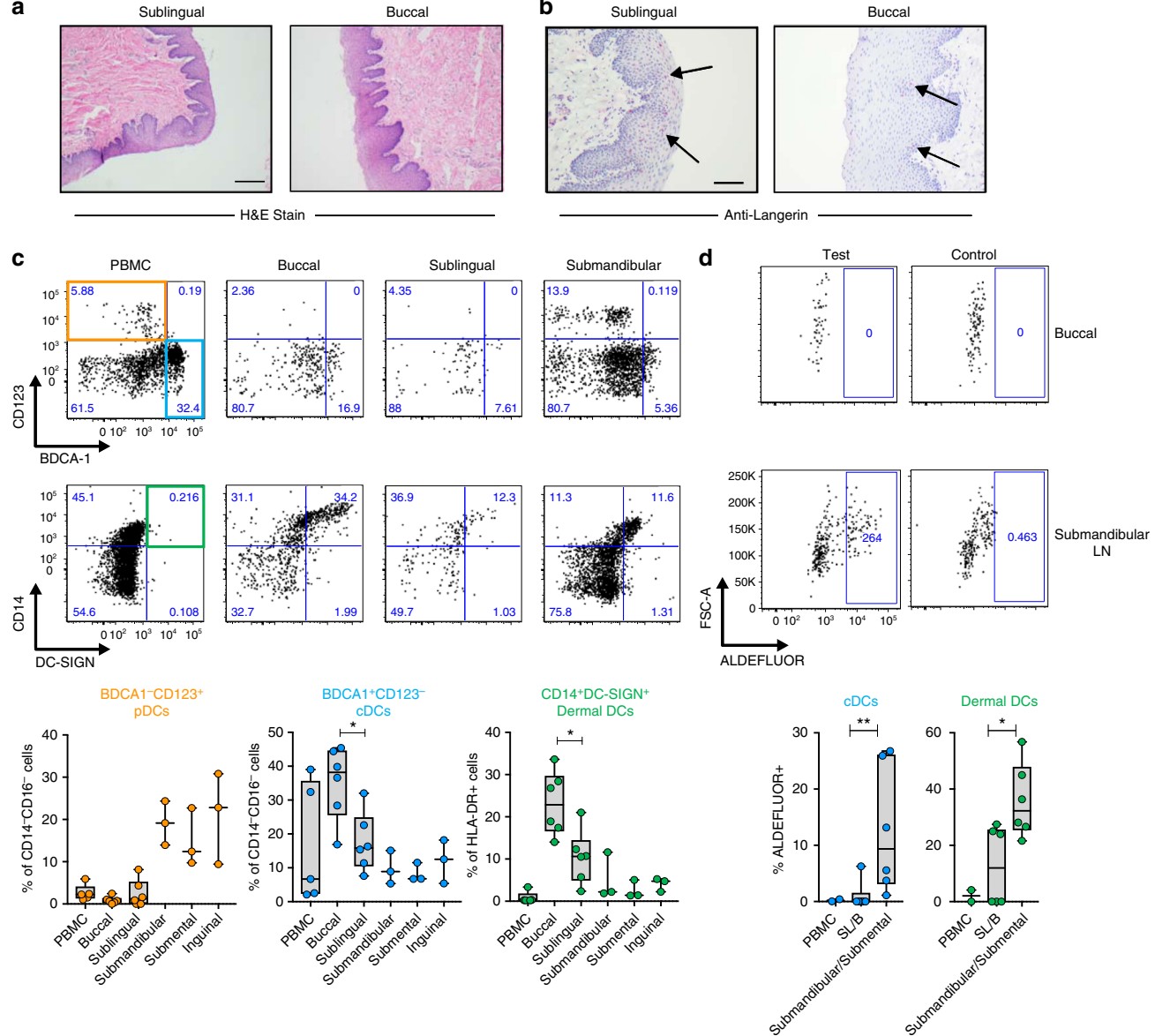

**Fig. 1** Dendritic cells in the sublingual and buccal (SL/B) Tissue of rhesus macaques. Paraffin-embedded sublingual and buccal tissue sections were stained with **a** Hematoxylin and eosin and imaged by light microscopy (scale bar, 500 μm) or **b** anti-Langerin and analyzed by immunohistochemistry (scale bar, 200 μm). Arrows show Langerin+ cells stained in red. **c** Representative flow plots showing dendritic cell (DC) subsets in PBMC, sublingual tissue, buccal tissue, and the submandibular, submental, and inguinal lymph nodes. Top row; CD45+CD3−CD20−HLA-DR+CD14−CD16− cells were gated for plasmacytoid DC (CD123+BDCA-1−, orange) and conventional DC (CD123−BDCA-1+, blue) markers. Middle row; CD45+CD3−CD20−HLA-DR+ cells were gated for Dermal DCs (CD14+DC-SIGN+, green). Bottom row; frequencies of DC subsets in different tissues (*, $p < 0.05$, Wilcoxon matched-pairs test). Data representative of samplings from six independent animals, $n = 5$ (PBMC) $n = 6$ (buccal, sublingual), $n = 3$ (submandibular, submental, and inguinal lymph nodes). Samples from the same animal were paired together. **d** Detection of aldehyde dehydrogenase (ALDH) activity in buccal or submandibular DCs using the ALDEFLUOR™ fluorescent reagent. Control samples were used to set gating. Representative flow plots are shown on top measuring ALDH activity in conventional DCs in buccal tissue compared to submandibular lymph nodes. Bottom, frequencies of ALDEFLUOR+ conventional DCs (blue) and dermal DCs (green) in PBMCs ($n = 2$), buccal ($n = 3$) and sublingual tissue ($n = 3$), and submandibular ($n = 3$) and submental lymph nodes ($n = 3$) (*, $p < 0.05$; **, $p < 0.01$; Mann–Whitney test). PMBC, peripheral blood mononuclear cells; SL/B, sublingual and buccal tissue. Box and whiskers plot in **c**, **d**; box extends from 25th to 75th percentile, the line indicates median, whiskers indicate min and max values

the submandibular and submental lymph nodes, however this activity was not found in DCs within the sublingual and buccal tissue themselves (Fig. 1d). However, it is possible that RALDH activity may be induced in tissue DC upon activation as we showed previously following Ad5 vaccination in mice[25]. Taken together, these data demonstrate that the oral mucosa contains multiple antigen-presenting cell subsets with predominance of LCs, BDCA1+ DC and CD14+DC-SIGN+ DC, and the draining lymph nodes contain gut-homing induction potential, indicating the oral mucosa is a viable route for mucosal vaccination.

**NHP study design.** To test if needle-free delivery of vaccines to the SL/B mucosa enhances the humoral immunity, we used the Syrijet Mark-II Needless Injector (Keystone Industries), designed for use in dentistry to inject local anesthesia to the oral mucosa[27] to deliver MVA-HIV and protein vaccines to rhesus macaques

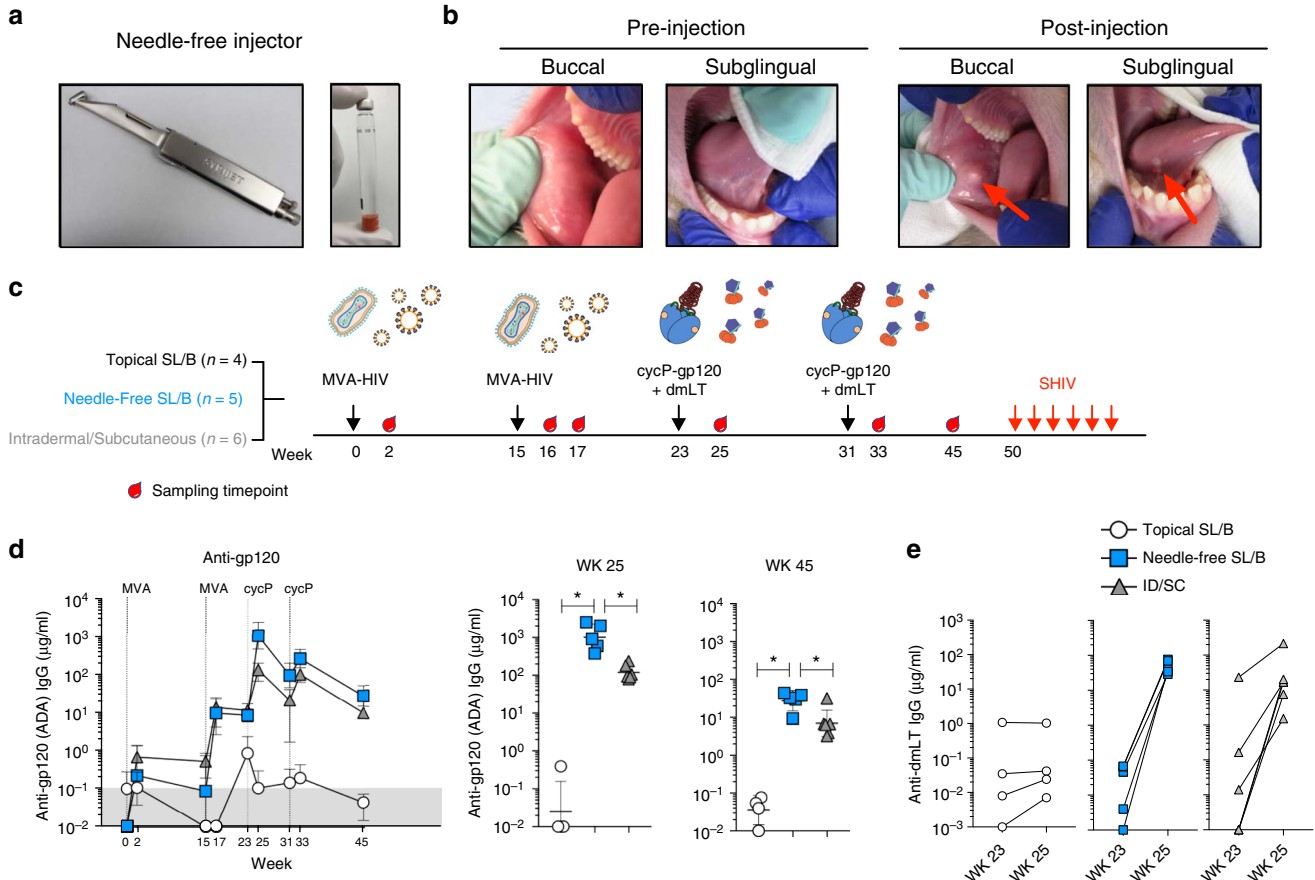

**Fig. 2** SL/B immunization with a needle-free injector induces strong systemic antibody responses. **a** Syrijet, the needle-free injector used to deliver immunizations to the sublingual and buccal tissue. Sterile water cartridges were modified to contain immunogens. **b** Sublingual and buccal tissue of a rhesus macaque before and five minutes after 100 μl PBS injection via Syrijet. **c** Study design. Rhesus macaques ($n = 15$) were immunized twice with MVA-HIV ($1 \times 10^8$ pfu) and boosted twice with recombinant trimeric gp120 (cycP-gp120) (100 μg) with the mucosal adjuvant dmLT. Animals were immunized via topical application to the sublingual and buccal (SL/B) tissue ($n = 4$), needle-free injection to the SL/B tissue ($n = 5$), or intradermally (with MVA-HIV) and subcutaneously (with cycP-gp120 + dmLT) ($n = 6$). MVA-HIV and cycP-gp120 doses were split between the buccal and sublingual tissue (SL/B) or the left and right thigh (ID/SC). 19 weeks following the second cycP-gp120 immunization, animals were challenged intra-rectally with low dose pathogenic SHIV-SF162P3 weekly for up to six weeks. Cartoons depict MVA-HIV and virus-like particles, cycP-gp120, and dmLT. **d** Kinetics of anti-gp120 (ADA) serum IgG in vaccine groups (geomean ± SD) and for individual animals (line, geomean ± SD) at the peak time point (wk 25) and pre-challenge time point (wk 45) (*, $p < 0.05$; Mann–Whitney test). Dotted lines denote week of indicated immunization. **e** Anti-dmLT serum IgG response in animals before (wk 25) and two weeks post (wk 45) immunization with cycP-gp120 + dmLT. **d, e** White circle, topical SL/B ($n = 4$); blue square, needle-free SL/B ($n = 5$); gray triangle, ID/SC ($n = 6$)

(Fig. 2a). First, we injected a rhesus macaque with 100 μl of sterile PBS and visualized the injection site before and after injection (Fig. 2b). The Syrijet injected effectively into both the sublingual and buccal tissue with minimal damage and bleeding, and no additional swelling or damage was reported in the following days post injection.

To evaluate the SL/B route of oral vaccination, we immunized female rhesus macaques orally via either topical application to the SL/B tissue ($n = 4$), needle-free injection to the SL/B tissue ($n = 5$), or the conventional intradermal and subcutaneous route (ID/SC) ($n = 6$). Macaques were immunized twice with $1 \times 10^8$ pfu (plaque-forming units) of MVA-HIV expressing HIV-1 clade B gag, pol, env (strain ADA) followed by two boosts with 100 μg of trimeric clade B cycP-gp120 immunogen (strain JRFL) along with the mucosal adjuvant dmLT (Fig. 2c)[13,28]. Doses were split in half between the sublingual and buccal tissue or the left and right thigh of ID/SC immunization.

MVA-HIV is a well-characterized viral vector for HIV-1 vaccines, currently being tested in human clinical trials as a potential vaccine candidate for HIV-1 and expresses trimeric HIV-1 envelope (gp150) on the surface of infected cells and produces virus-like particles (VLPs)[15,29]. CycP-gp120 is a novel trimeric gp120 immunogen, stabilized by a heterologous trimerization domain inserted into the V1-loop of a cyclic permutated gp120[18,19]. Based on previous dose-escalating studies evaluating dmLT as a mucosal adjuvant, needle-free SL/B and topical SL/B cycP-gp120 immunization were adjuvanted with 25 μg of dmLT per site, while subcutaneous cycP-gp120 immunization in the ID/SC group was adjuvanted with 1 μg of dmLT. Since we observed swelling of mouth following the first cycP-gp120 + dmLT boost, and to make the second boost comparable between needle-free SL/B and ID/SC immunization, we reduced the dose of dmLT for the second cycP-gp120 immunization to 1 μg per site[30].

**Needle-free vaccination induces strong serum IgG responses.** Impressively, needle-free SL/B immunization generated a strong anti-ADA gp120 IgG response in serum, a feature that is not common for oral immunizations (Fig. 2d). Two MVA-HIV immunizations induced low levels of binding antibodies (geo-

mean titer, 9.6 µg ml⁻¹) that were boosted remarkably by 100-fold following the 1st protein boost (geo-mean titer, 1012 µg ml⁻¹). These responses contracted by 10-fold over 8 weeks and were marginally boosted upon the 2nd protein boost (geo-mean titer, 258 µg ml⁻¹). At the time of pre-challenge, 14 weeks after the final protein boost, anti-gp120 serum IgG titers had contracted about 10-fold (geo-mean titer, 27 µg ml⁻¹). In contrast, topical SL/B immunization resulted in minimal or undetectable antibody responses following both MVA-HIV and cycP-gp120 immunizations. Remarkably, while gp120-specific IgG responses induced by the needle-free SL/B immunization were comparable with ID/SC group following MVA immunizations, they were 10-fold (geo-mean titer 1012 vs. 120 ug/ml) and 3-fold higher (geo-mean titer, 258 vs 94 µg ml⁻¹) following the 1st and 2nd protein boosts, respectively. However, we are uncertain if this was because of the higher dose of the adjuvant used in the needle-free group during the 1st protein boost. Despite differences in the adjuvant doses, these data show that needle-free SL/B injection is an effective non-invasive method of generating high titers of vaccine-specific serum IgG. As dmLT is currently being investigated for its potential as a vaccine candidate against Enterotoxigenic Escherichia coli (ETEC) infection[31], we also measured serum IgG responses against dmLT and found a strong anti-dmLT IgG response generated in needle-free SL/B immunized animals upon boosting with cycP-gp120 adjuvanted with dmLT (Fig. 2e). These results demonstrate that needle-free SL/B immunization is an effective route to non-invasively generate strong systemic vaccine-specific IgG antibody responses against multiple antigens.

**Needle-free vaccination induces strong mucosal IgG and IgA.** As a major goal of mucosal vaccination is the generation of mucosal immune responses, we measured vaccine-specific IgG and IgA antibodies in the rectal, vaginal, and salivary secretions. Due to the variable amounts of immunoglobulin in secretions, antibody concentrations were normalized relative to the total IgG or IgA concentrations by calculating the specific activity (SA; ng gp120-specific IgG or IgA per total µg IgG or IgA)[32]. As with the systemic responses, needle-free SL/B immunization generated a strong gp120-specific IgG response in rectal, vaginal, and salivary secretions (Fig. 3a, b, c). Similar to serum IgG responses, the mucosal IgG responses also peaked after the 1st protein boost and the 2nd protein boost showed a small recall (<2-fold). At the peak (wk 25), the IgG SA was comparable between the three mucosal compartments (geo-mean of specific activity; rectal, 56; vaginal 43; salivary, 51). Topical SL/B immunization generated minimal to undetectable mucosal antibodies, highlighting the importance of the needle-free injector to generate these responses. Additionally, while ID/SC immunization did result in mucosal IgG responses, these were significantly lower than needle-free SL/B responses at the peak time point (wk 25). Mucosal IgG responses after the first protein boost contracted and were modestly expanded by the subsequent boost before contraction to the pre-challenge time point. Interestingly, while the gp120-specific rectal and vaginal IgG antibodies were maintained until pre-challenge, IgG antibodies in the saliva declined and approached undetectable levels, suggesting the establishment and maintenance of vaccine-specific IgG antibody responses varied between mucosal compartments.

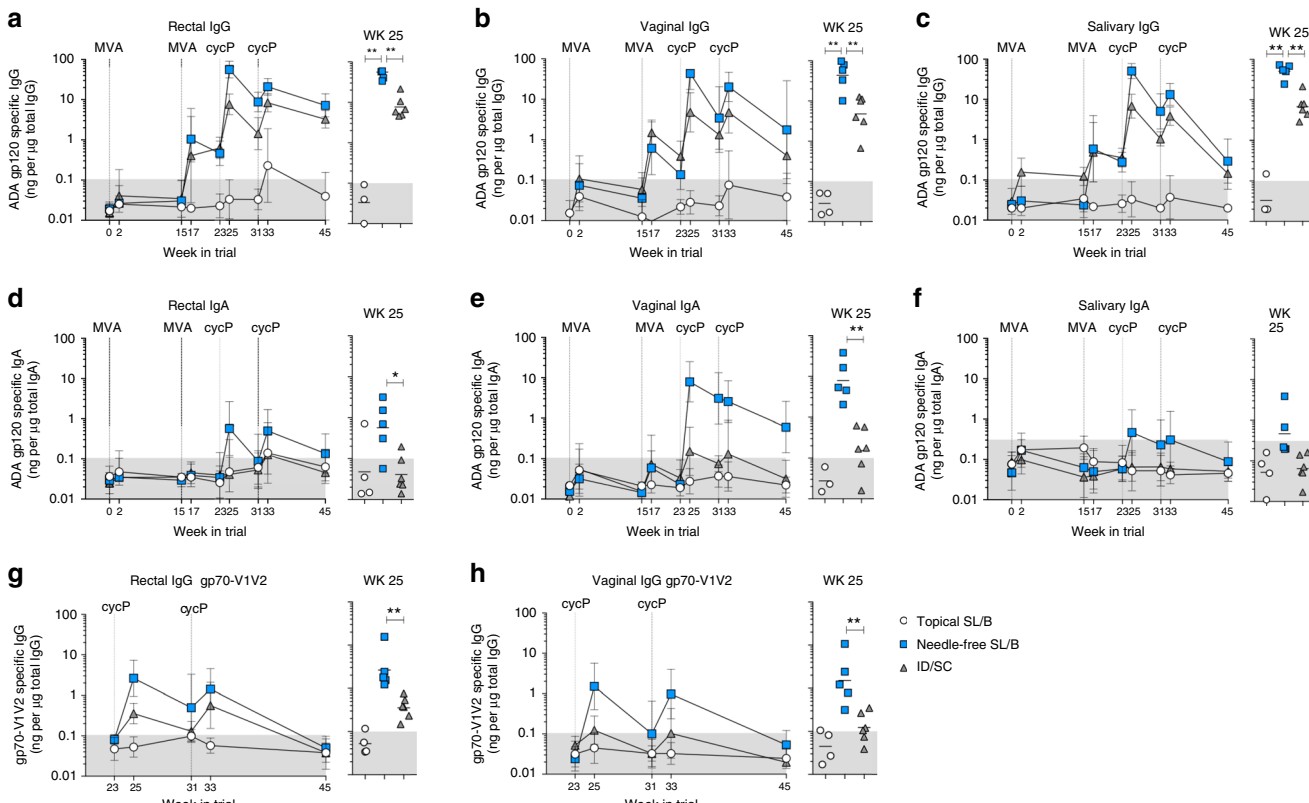

**Fig. 3** Needle-free SL/B immunization generates strong mucosal antibody responses. **a–c** Anti-ADA gp120 IgG antibodies in rectal (**a**), vaginal (**b**), and salivary (**c**) secretions. **d–f** Anti-ADA gp120 IgA antibodies in rectal (**d**), vaginal (**e**), and salivary (**f**) secretions. Data represented as geomean ± SD specific activity for each group. Specific activity calculated as ng gp120-specific IgG or IgA antibody per µg total IgG or IgA isolated. Specific activity for individual animals is shown at the peak time point (wk 25); line indicates geomean. The shaded region in each graph indicates the specific activity cut-off value. **g–h** IgG specific activity against gp70-V1V2 (Clade B/Case A2) in rectal and vaginal secretions. **a–h** * p < 0.05; ** p < 0.01; Mann–Whitney test. White circle, topical SL/B (n = 4); blue square, needle-free SL/B (n = 5); gray triangle, ID/SC (n = 6)

Needle-free SL/B immunization also generated vaccine-specific IgA antibodies in all three mucosal compartments, with strongest responses in vaginal secretions with all animals generating detectable vaginal IgA responses (geo-mean SA of 7.9) and weakest in saliva with only 2 out of 5 animals generating detectable IgA (Fig. 3d, e, f). The peak rectal and vaginal IgA response (wk 25) was significantly higher in needle-free SL/B compared to ID/SC immunized animals, which generated minimal mucosal IgA responses, and these responses were largely undetectable in topical SL/B immunized animals. Rectal IgA and salivary IgA levels contracted over time and were predominately undetectable at the time of pre-challenge. However, vaginal IgA responses were still detectable at the time of pre-challenge, suggesting this route may generate durable vaginal antibody responses. In addition to gp120 specific antibodies, we also measured antibodies against the gp70-V1V2 scaffold antigen, which displays the variable loops 1-2 (V1V2) loops of gp120, as serum IgG antibodies directed against gp70-V1V2 were a major correlate of protection in the RV144 trial[33,34]. Both needle-free SL/B and ID/SC immunization generated anti-gp70-V1V2 IgG in both rectal and vaginal secretions throughout the immunization regimen, however these responses were not long-lasting as they had contracted to undetectable levels by the pre-challenge time point (Fig. 3g, h). These results indicate that needle-free SL/B immunization is an easy and practical method of generating strong IgG responses in rectal, vaginal and oral mucosa, and strong IgA responses in vaginal mucosa.

**Needle-free vaccination induces broad V1V2 and V2HS response**. To address the global diversity of HIV-1, an effective vaccine should ideally recognize HIV-1 isolates from multiple strains and clades. To characterize the cross-reactivity of MVA/cycP-gp120 induced antibodies, we measured antibody binding to a global panel of gp120, gp140 and gp70-V1V2 scaffold proteins via binding antibody multiplex assay (BAMA)[35]. Upon boosting with cycP-gp120 we observed a strong cross-reactive antibody response against multiple clades of gp120 and gp140 antigens, reacted to by all immunized animals (Fig. 4a). Responses were significantly higher in 15/16 of the antigens tested for needle-free orally immunized animals compared to ID/SC immunized at this time point (wk 25), however at the pre-challenge time point both groups had similar levels of reactivity (Supplementary Fig. 2a). These results show the high broadly-reactive antibody response generated by MVA/cycP-gp120 regimen, a crucial component to an HIV-1 vaccine candidate.

As results from the RV144 trial suggest that antibodies directed towards the V1V2 loop of gp120 are associated with reduced risk of infection[33], the generation of these antibodies, especially broadly reactive V1V2-directed antibodies, is of great interest in HIV-1 vaccine development. To measure the cross-reactivity of V1V2-directed antibodies, sera IgG binding to a panel of 16 gp70-V1V2 scaffolds representing the global diversity of HIV-1 was quantified via BAMA. Two weeks post the first protein boost, needle-free SL/B immunized animals generated a substantial broadly cross-reactive gp70-V1V2 response against multiple clades of isolates, significantly higher than ID/SC immunization, demonstrating not only the high immunogenicity of the sublingual and buccal route, but also the broadening of the antibody responses upon needle-free oral delivery (Fig. 4b). Similarly, the anti-Env response at the pre-challenge time point responses to gp70-V1V2 scaffolds had contracted in both groups, with no significant differences between the groups, suggesting that through contraction and further boosting in ID/SC immunized animals the V1V2 response leveled to a set point (Supplementary Fig. 2b).

To map the regions of gp120 targeted by the vaccine-induced antibody response, we measured binding via peptide microarray of sera to 15-mer peptides (overlapping by 12 amino acids) derived from 13 strains including consensus clade B Env (Supplementary Table 1). Both needle-free SL/B and ID/SC immunization resulted in a broad response against numerous regions of consensus clade B gp120, the strongest responses directed against the C1, C2, V3, and C5 regions of gp120 (Fig. 4c). Binding responses against linear V2 hotspot epitope[36] was developed against consensus B (Fig. 4c) as well as to other clade consensus and viral strains at lower magnitude (Supplementary Fig. 2c). Linear V2 binding was a subdominant response compared to V3 and C4 linear epitope binding, consistent with previous findings for HIV-1 Env vaccine-elicited antibody responses[37]. Comparing needle-free SL/B to ID/SC peptide responses showed similar trends in both groups, with a high proportion of the IgG response directed towards the C1, V3, and C5 regions, however needle-free SL/B immunization resulted in a modestly larger proportions of response against the V2, C4, V5, and C5.1 regions (Fig. 4d).

We next measured antibody responses to the V2 Hotspot (V2-HS) peptide, a region of the V2 loop (spanning positions 166–178 of HIV-1 stain HXB2) in which antibody recognition correlated significantly with decreased risk of infection in the RV144 trial[36,38]. Responses to the consensus B V2 peptides were detected in the peptide microarray analysis, primarily in the needle-free SL/B group, though these responses were modest compared to other regions. To determine responses to the V2-region of the vaccine and challenge virus strains, we synthesized 13 amino-acid peptides corresponding to the V2 hotspot of cycP-gp120 (clade-B JRFL, E168K), MVA-HIV (clade-B ADA), and SHIV-SF162P3 (clade-B)[39]. Needle-free SL/B immunization generated a strong cross-reactive V2 hotspot response, recognizing not only the MVA-HIV and cycP-gp120 vaccine strains (ADA, JRFL E168K), but also the heterologous SHIV-SF162P3 (Fig. 4e), similar to previous studies examining antibody responses generated by cycP-gp120[19]. ID/SC immunization resulted in minimal V2 responses compared to needle-free SL/B immunization, which is likely due to the overall antibody response being significantly lower in ID/SC immunized animals at this time point. The cross-reactivity of V2 antibody responses to heterologous strains detected in both linear peptide microarray and in V2 hotspot ELISA demonstrates the remarkable ability of cycP-gp120 to generate V2-hotspot binding antibodies.

The generation of neutralizing antibodies to HIV-1 is a long-sought goal of HIV-1 vaccination. To test the presence of neutralizing antibodies generated by MVA-HIV/cycP-gp120 immunization, we measured the neutralizing activity of sera against a multi-clade panel of pseudoviruses that have high (SF162.LS, MW965.26), moderate (BaL.26), or low (ADA, JR-FL, TRO.11) neutralization sensitivity (tier-1A, -1B, -2, respectively). Needle-free SL/B immunization induced a significantly higher titer of neutralizing antibodies against the neutralization sensitive clade-B SF162.LS and the moderately resistant clade-B BaL.26 isolate after the first and second protein boosts than ID/SC immunization (Fig. 4f). At the pre-challenge time point, however, both responses were not significantly different. Additionally, we detected no tier-2 (neutralization resistant) neutralizing antibody responses, which is in line with our previous results[19], suggesting that mutations to cycP-gp120, or additional homologous immunizations[18], may be required to generate tier-2 neutralizing activity.

The results of the RV144 trial suggest that non-neutralizing antibodies may have a role in protection as vaccine-induced V1V2-directed antibodies were largely non-neutralizing[33]. Furthermore, non-neutralizing antibodies have been shown to

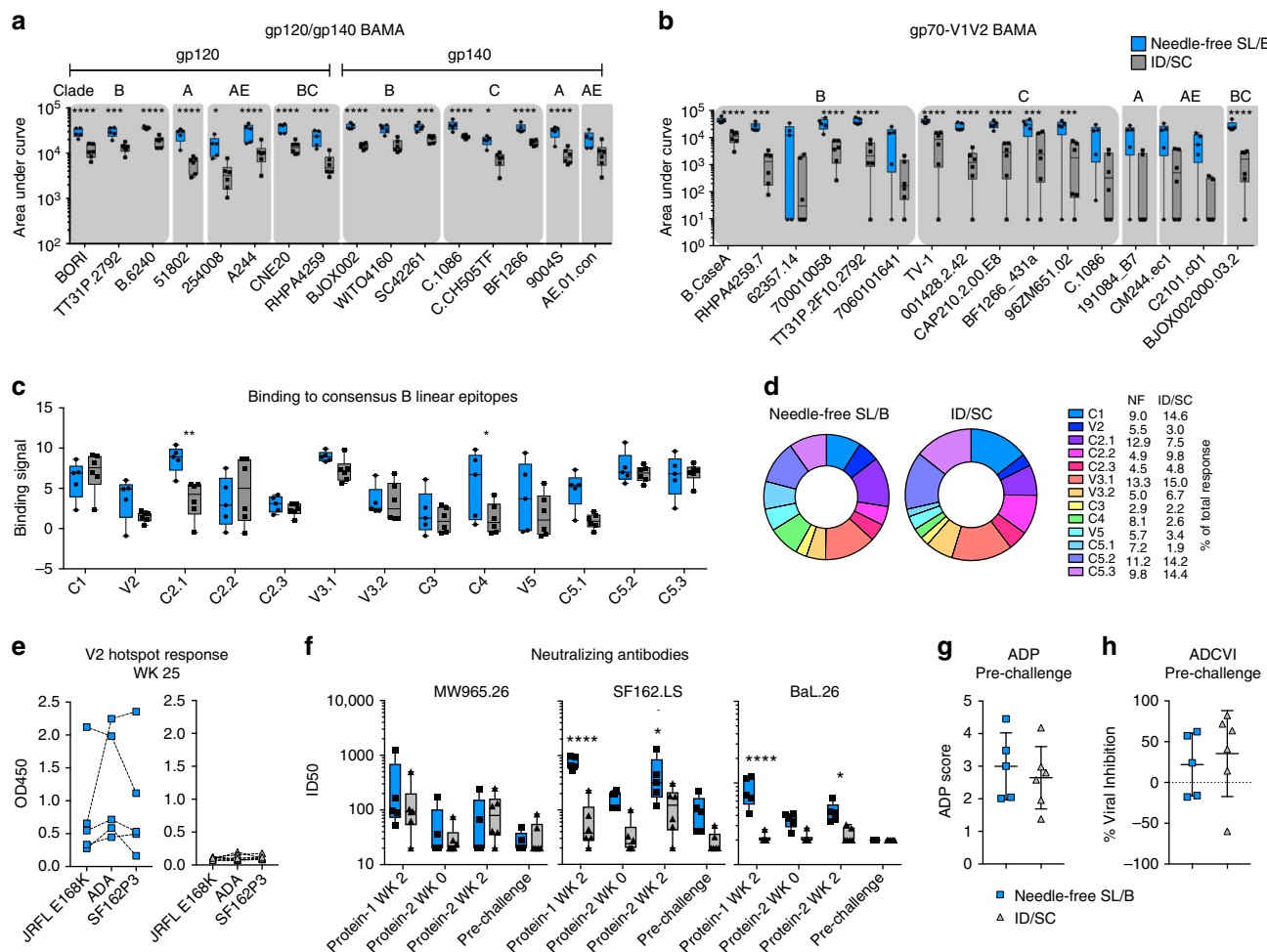

**Fig. 4** Serum IgG specificity, neutralizing activity, and effector function. Binding of IgG antibodies at two weeks post the first cycP-gp120 immunization (wk 25) to **a** gp120 and gp140 antigens and **b** gp70-V1V2 scaffolds representing the global diversity of HIV-1 determined using Binding Antibody Multiplex Assay (BAMA). Shaded areas indicate clade. **c** Binding of peak immune sera (wk 25) to 15mer peptides derived from clade B consensus gp120, measured by peptide microarray linear epitope mapping and reported as binding signal (Log2 fold difference post-immunization/baseline binding intensity). Magnitude of binding to each epitope is defined as the highest binding signal for a single peptide within the region of the epitope. **d** Representation of peptide array binding of each clade B consensus epitope as % of the total response. **e** Anti-V2 hotspot (HS) response (wk 25) against V2-HS peptides derived from strains JRFL (E168K), ADA, and SF162P3 measured by ELISA. Dotted lines connect data from same animal. Blue squares, needle-free SL/B; gray triangles, ID/SC. **f** Neutralizing antibodies over time against HIV-1 isolates MW965.26, SF162.LS, and BaL.26, measured as ID50. **g** Antibody-dependent phagocytosis (ADP) at pre-challenge (wk 45). ADP score (mean ± S.D.) calculated for each serum by dividing the median fluorescence intensity (MFI) of bead positive cells by the value obtained using the same dilution of pooled serum from naive macaques. **h** Antibody-dependent cell viral inhibition (ADCVI) measured at pre-challenge as % viral inhibition (mean ± S.D.). An average of two individual experiments is shown. Box and whiskers plots (**a–c**, **f**); box extends from 25th to 75th percentile, line indicates median, whiskers indicate min and max values (*$p < 0.05$; **$p < 0.01$; ***$p < 0.001$; ****, $p < 0.0001$; 2-Way ANOVA, multiple comparisons). ID50, serum dilution required to neutralize 50% infection

be a correlate of protection from HIV, SIV, and SHIV infection in both human and non-human primate studies[40]. To evaluate non-neutralizing antibody effector functionality, we characterized antibody-dependent phagocytosis (ADP) (Fig. 4g), which measures the internalization of gp120-coated by monocytes via the binding of anti-gp120 IgG to Fc receptors, and antibody-dependent cellular viral inhibition (ADCVI) (Fig. 4h), which measures the combined ability of monocytes and natural killer (NK) cells (within human PBMCs) to eliminate infected target cells (HIV SF162-infected CCR5+ CEM-NKr cells) and released cell-free virus in the presence of Env-specific antibodies. In serum, taken at pre-challenge, we measured a range of ADP and ADCVI activity in both vaccine groups, suggesting a breadth of effector functionality in the antibody response. Interestingly, when measuring ADCVI activity, we found that sera from some

animals enhanced viral outgrowth rather than inhibiting replication, whereas other sera inhibited up to 80% of viral outgrowth compared to controls (effectors + targets + naive serum). ADCVI activity was not related to the magnitude of the antibody response at the pre-challenge time point (Supplementary Fig. 2d), indicating that antibody effector functionality is more dependent on specificity than magnitude, and certain antibody responses may be detrimental in combating HIV-1.

**Needle-free vaccination induces T cell responses in blood**. To measure the cellular immune responses after immunization, we stimulated peripheral blood mononuclear cells (PBMCs) with HIV-1 clade B consensus Gag and Env peptides and measured cytokine production in T-cells (Live CD3+ cells) by flow cytometry (Supplementary Figure 3a, Fig. 5a). Immunization with

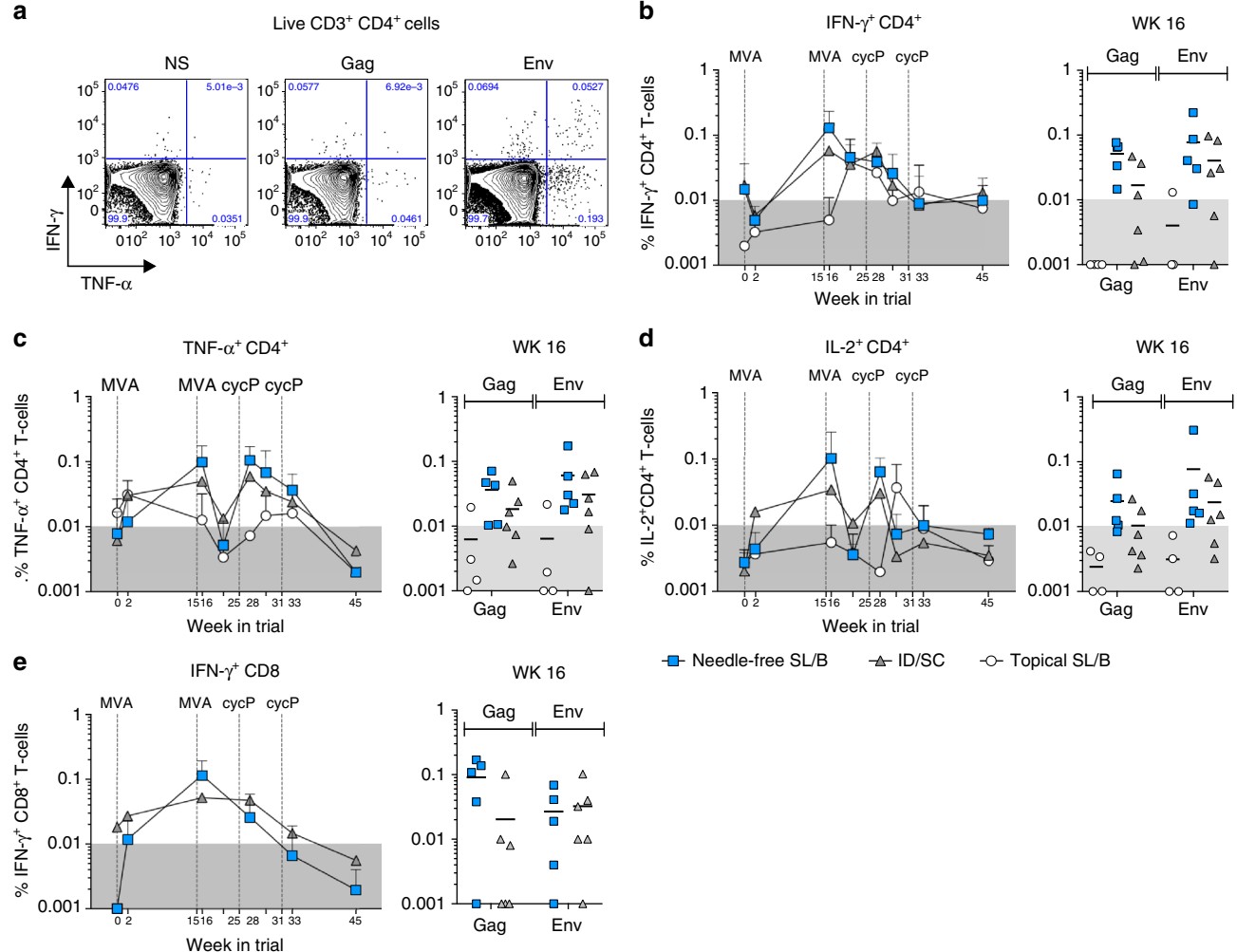

**Fig. 5** Needle-free SL/B immunization generates vaccine-specific CD4 and CD8 T cells in the blood. PMBCs were stimulated with HIV-1 consensus B Gag and Env peptides and analyzed by flow cytometry for cytokine production. **a** Representative flow plots for IFN-γ and TNF-α cytokine expression on Live CD3+CD4+ cells in non-stimulated (NS), Gag, or Env stimulated PBMCs is shown. Kinetics of the total (Gag + Env) **b** IFN-γ, **c** TNF-α, and **d** IL-2 response in CD4+ T cells (mean ± S.D.), with the peak response (wk 16) highlighted for each animal (line denotes mean). **e** Kinetics of the total IFN-γ response in CD8+ cells (mean ± S.D.), with the peak response (wk 16) highlighted for each animal. White circle, topical SL/B ($n = 4$); blue square, needle-free SL/B ($n = 5$); gray triangle, ID/SC ($n = 6$)

MVA-HIV resulted in IFN-γ production in CD4+ T-cells against both Gag and Env peptides, peaking one week after the second MVA-HIV immunization (wk 16), and responses were similar between needle-free SL/B and ID/SC groups (Fig. 5b). Similar to IFN-γ responses, Gag and Env specific TNF-α (Fig. 5c) and IL-2 (Fig. 5d) producing CD4+ T-cells were also observed, peaking again after the second MVA-HIV. Interestingly, unlike IFN-γ production, both TNF-α and IL-2 responses were expanded upon the first cycP-gp120 + dmLT boost. In contrast to vaccine-specific CD4 T-cell responses, we detected only low levels of vaccine-specific CD8 T-cell responses that peaked after the second MVA-HIV immunization and contracted to below detection over the course of the immunization (Fig. 5e). However, we did not detect vaccine-specific T-cell responses in the rectal or vaginal tissues after immunization, suggesting that stronger T-cell generating vaccine strategies, such as the utilization of DNA-primes, may be necessary to induce these responses[41]. As a major concern with HIV-1 vaccines is the unwanted generation of an abundance of HIV target CD4+ T-cells in the mucosal tissue, we characterized the phenotypes of CD4 T-cells in rectal tissue at time of pre-challenge and found that needle-free SL/B, topical SL/B, and ID/SC immunized animals all had similar levels of activated

(HLA-DR+CCR5+) CD4+ T-cells in the rectal tissue, indicating that immunization did not result in an accumulation of target cells in the rectal mucosa (Supplementary Fig. 3b,c). Taken together, these data show that needle-free SL/B immunization is capable of generating vaccine-specific functional T-cell responses in the blood, similar to responses generated by conventional ID/SC immunization.

**MVA-HIV/cycP-gp120 vaccination protects from SHIV challenge.** As a preliminary readout of the efficacy of MVA-HIV/cycP-gp120 vaccine, we challenged animals 19 weeks after the second protein boost with repeat low dose weekly intrarectal challenges of SHIV-SF162P3 for maximum of six exposures[39]. The envelope in SHIV-SF162P3 virus is a tier-2 Env, heterologous to the vaccine strains, and we measured no detectable neutralization against this Env in vaccinated animals. Five unvaccinated female macaques were used as a control group. Following challenge, all unvaccinated animals were infected by the third challenge with 3 of the 5 becoming SHIV infected after the 1st exposure. Impressively, we detected a significant delay in acquisition of infection in both the needle-free SL/B immunized

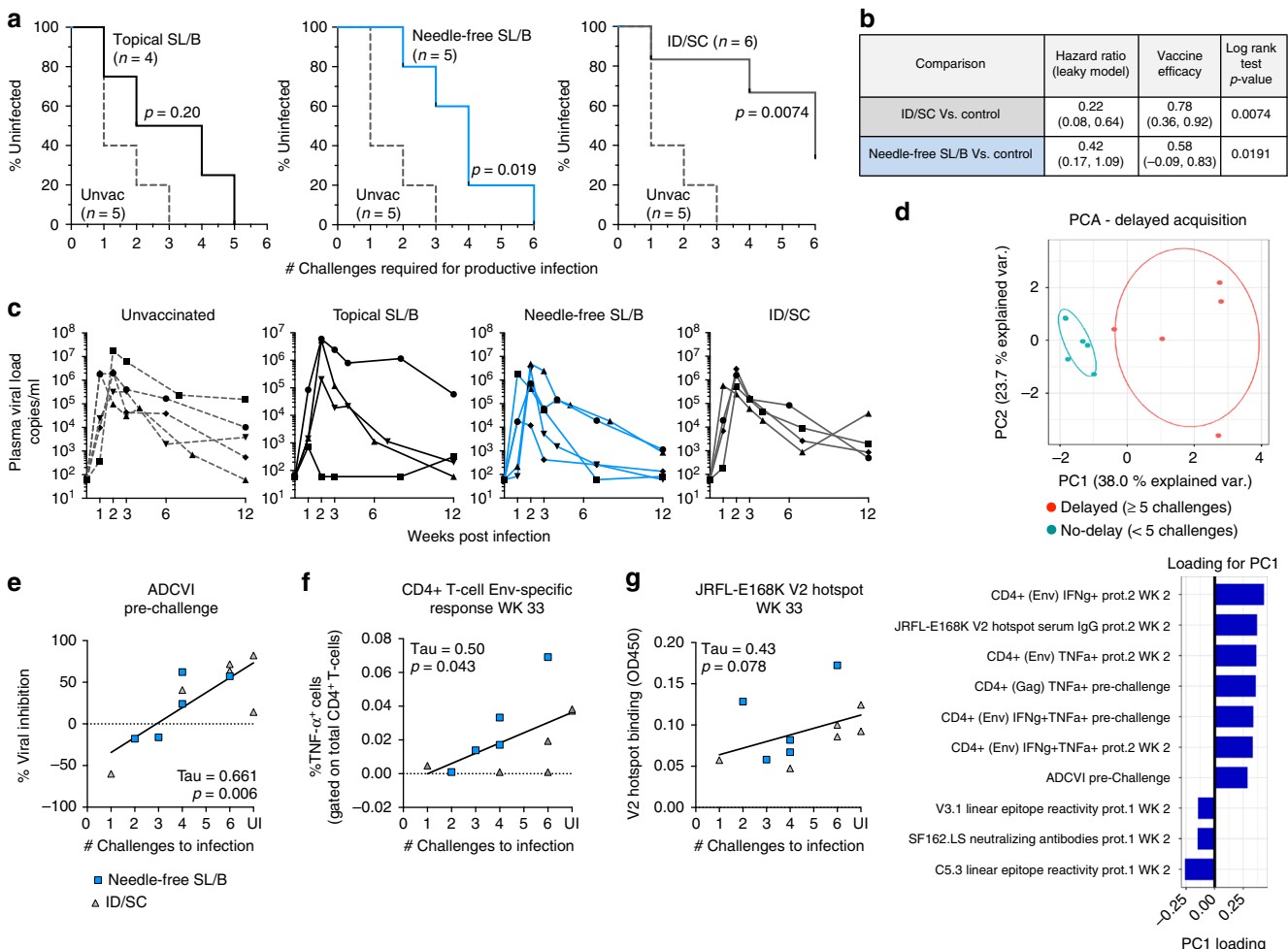

**Fig. 6** Needle-Free SL/B and ID/SC immunization results in delayed acquisition of SHIV-SF162P3 infection. Animals were challenged weekly with an intra-rectal low dose (1:100 dilution) of SHIV-SF162P3. 5 unvaccinated macaques were used as a control. **a** Acquisition of infection in topical SL/B, needle-free SL/B, or ID/SC immunized animals (Log-rank (Mantel-Cox) test). Dotted line, unvaccinated controls ($n = 5$); black line, topical SL/B ($n = 4$); blue line, needle-free SL/B ($n = 5$); gray line, ID/SC ($n = 6$). **b** Vaccine efficacy of ID/SC, needle-free SL/B immunization groups vs control animals (Log-rank (Mantel-Cox) test). **c** Kinetics of plasma viral loads in unvaccinated and vaccinated animals. **d** Top, Principal Comparison Analysis (PCA) plot showing PC1 and PC2 scores for delayed ($\geq 5$ challenges to infect, $n = 5$) or non-delayed (<5 challenges to infect, $n = 6$) acquisition of infection in needle-free SL/B and ID/SC immunized animals. Bottom, loadings of immune response parameters in PC1. **e-g** Correlation analysis of ADCVI activity at pre-challenge, Env-specific CD4$^+$ T cell TNF-$\alpha$ production two weeks after the second cycP-gp120 immunization (wk 33), and JRFL-E168K V2 hotspot reactivity in serum IgG (wk 33) with acquisition of infection. (Kendall's Tau Correlation Test). Needle-free SL/B (blue squares) and ID/SC (gray triangles) are combined for analysis ($n = 11$)

($p = 0.019$) and ID/SC ($p = 0.0074$) immunized animals, but not topical SL/B immunization ($p = 0.20$) (Log-rank (Mantel-Cox) test), with two ID/SC immunized animals remaining uninfected after the sixth challenge (Fig. 6a). Vaccine efficacy per exposure for each group was calculated to be 58% (Needle-Free SL/B) and 77% (ID/SC) (Fig. 6b). No differences in vaccine efficacy were observed between these two vaccine groups. Tracking viral loads following infection, we detected a trend for lower set-point viral loads in immunized animals, but these did not reach significance (Fig. 6c).

Due to the low sample size in the vaccine groups, elucidating clear correlates of protection can be challenging. As both needle-free SL/B and ID/SC showed a significant delay in acquisition, as well as receiving the same vaccine immunogens, we combined the two groups for correlation analyses. To identify immune response profiles that could differentiate animals that showed delayed acquisition or were protected (>/=5 challenges) versus no delay in acquisition (<5 challenges), we performed a principal component analysis (PCA) with the all immune responses measured in this study. PCA results showed that principal

component 1 (PC1), containing ten variables, could separate animals with and without delayed acquisition (Fig. 6d), while a simulated PCA performed with the ten PC1 variables using randomized assay data showed no separation (Supplementary Fig. 4a). Loadings for PC1 included CD4$^+$ Env specific T cell responses, pre-challenge ADCVI activity, and V2 hotspot binding, as well as lower C5.3 linear epitope binding (Supplementary Fig. 4b). Univariate analysis for these variables confirmed a significant positive correlation between the acquisition of infection and pre-challenge ADCVI activity (Fig. 6e), Env-specific TNF-$\alpha^+$ CD4 T cells (Fig. 6f) or Env-specific IFN-$\gamma^+$ TNF-$\alpha^+$ cells (Supplementary Fig. 5a). In addition, there was a trend for positive correlation between the acquisition of infection and V2 hotspot binding (Fig. 6g). In addition, several parameters of the immune response correlated or highly trended with protection from infection, but these were observed only within single groups. These include the peak (wk 25) serum IgG response, rectal IgG responses at pre-challenge, and the V2 hotspot response against the SHIV-SF162P3 challenge virus

strain (Supplementary Fig. 5b). Taken together, these data show that MVA-HIV/cycP-gp120 induced T-cell and antibody responses contribute to protection against pathogenic SHIV-SF162P3 infection.

## Discussion

The oral mucosa has been studied extensively as a site of mucosal vaccination in animal models predominately through the topical application and natural uptake of vaccines by the oral mucosa[6,42]. In this study, we tested the influence of delivering vaccines across the mucosa using a needle-free device on generating a strong mucosal and systemic immunity. Our results demonstrated that needle-free injection of the sublingual and buccal tissue is an easy, safe, and efficient method of vaccination for generating strong mucosal and systemic antibody responses. Needle-free SL/B immunization also resulted in induction of strong CD4 and CD8 T cell responses in blood that were comparable to systemic immunization. The strong immunogenicity of the SL/B tissue, which is often regarded as a site for immune tolerance induction[6,43], suggests the adjuvant activity of MVA-HIV and dmLT effectively overcomes the tolerogenic nature of the oral tissue to generate vaccine-specific T and B cell responses. The concept of needle-free delivery of vaccines is not new as recent studies examined needle-free injection through the skin as a method of vaccination in humans[10,11,44–46]. However, to our knowledge this method of vaccination has not been explored for SL/B route in humans or non-human primates and thus our study represents the first demonstration that needle-free SL/B immunization as an attractive approach for mucosal vaccination with protective potential against HIV, warranting further studies to characterize this route for HIV/SIV and other mucosal pathogens.

An impressive finding of our study is that oral needle-free vaccination induces strong humoral and cellular responses in blood in addition to mucosal compartments. This was in contrast to previous oral topical vaccination studies, which showed relatively poor systemic immunity[47,48] highlighting the unique ability of needle-free vaccination to generate systemic immunity. One question that remains to be addressed is how comparable are the systemic immune responses induced by the oral needle-free vaccination to systemic immunization. We are unable to determine this in this study due to the use of higher adjuvant dose during the 1st protein boost. Nevertheless, our data show that at least after MVA vaccinations they are comparable.

The mechanisms by which needle-free SL/B vaccination induces a strong mucosal and systemic immunity need further investigation. In addition, since we vaccinated via both buccal and sublingual routes simultaneously, we cannot conclude if buccal or sublingual vaccination alone is superior compared to each other. To gain insights into the role of different antigen-presenting cells in the mouth, we characterized different subsets of DC in the buccal and sublingual tissue. Our results showed the presence of multiple subsets of DCs, including Langerhans cells (LCs), CD14$^+$DC-SIGN$^+$ dermal DCs, and BDCA-1$^+$ (CD1c$^+$) conventional DCs (cDCs). These subsets have been shown to have multiple roles in the generation of tolerance or immunity. For example, LCs are generally considered to be tolerogenic and aid in the maintenance of the epithelial barrier, however there is increasing evidence that these cells have some functional plasticity and are capable of becoming immunogenic based on the specific context of inflammation[49,50]. In contrast, cDCs are thought to be more involved with the induction of immunity and cross-presentation of antigens, and dermal DCs or similar CD14$^+$ cells have been shown to be important in T-follicular helper cell formation and B-cell activation[8,20,51]. Additionally, while these subsets were found in both buccal and sublingual tissue, we found that

the buccal tissue had significantly higher proportion of both BDCA-1$^+$ cDCs and dermal DCs than the sublingual tissue. The larger proportion of these subsets suggests the buccal tissue may be a preferred target of immunization, although most oral tissue immunization studies have targeted the sublingual tissue[6]. Future studies should evaluate buccal vs sublingual only immunization to compare the differences in these tissues, as well as evaluate the effect different adjuvants have on the activation and function of the different dendritic cell subsets.

It is well known that DCs in the gut-associated LNs express high levels of RALDH enzymes, which are important to convert vitamin A into the gut-homing imprinting retinoic acid. Interestingly, we did not detect strong RALDH activity in buccal and sublingual DC subsets. However, we did detect this activity in DCs within the oral draining lymph nodes, demonstrating these lymph node-resident DC are capable of imprinting gut-homing characteristics on T- and B-cells[23]. It is possible that DCs in the buccal and sublingual tissue upregulate RALDH activity upon activation by MVA as we showed previously for Ad5 vaccination in mice[25], or traffic vaccine antigens to the lymph node in which resident DCs imprint gut-homing characteristics. Further studies elucidating which DCs take up the antigen and present to T and B cells in the SL/B mucosa will help to understand the mechanisms of mucosal priming and further improving the effectiveness of this route in generating vaccine responses.

This study also demonstrates for the first-time protective efficacy of MVA-HIV/cycP-gp120 vaccination against a heterologous tier-2 neutralization resistant SHIV-SF162P3 infection in macaques[52]. MVA-HIV/cycP-gp120 immunization induced broadly cross-reactive gp120, gp140, and gp70-V1V2 serum IgG, tier-1A and -1B neutralizing antibodies, gp120-specific antibody-dependent phagocytic (ADP) activity, antibody-dependent cell-mediated viral inhibition (ADCVI) activity, and vaccine-specific CD4$^+$ and CD8$^+$ T-cell responses in PBMCs. Upon challenging immunized animals intra-rectally with a low dose of heterologous, hard to neutralize, SHIV-SF16P3, we observed a delayed acquisition of infection in both needle-free SL/B and ID/SC vaccine groups, compared to unvaccinated controls. Topical SL/B immunized animals showed no significant protection. The lack of neutralizing antibodies against tier-2 neutralization resistant HIV-1 isolates, including against SHIV-SF162P3, indicates non-neutralizing antibody effector functions may be important for protection[53]. Furthermore, while tier-1A or -1B neutralizing antibodies did not correlate with protection, we observed a significant correlation in ADCVI activity at pre-challenge in needle-free SL/B and ID/SC immunized animals. ADCVI activity is a measurement of a combination of anti-viral antibody effector functions, including ADP, opsonization, and antibody-dependent cellular cytotoxicity (ADCC), which may synergize in inhibiting viral replication. Interestingly, ADP activity alone did not correlate with protection (Supplementary Fig. 5c), suggesting additional anti-viral functions measured by ADCVI are required for protection. In addition to ADCVI activity, we found a significant correlation with anti-Env CD4+ T-cell response after the second cycP-gp120 immunization, as well as a positive trend in anti-V2 antibodies with delayed SHIV acquisition, supporting recent studies highlighting the importance of V2-specific antibodies in the human efficacy trials[34,38,54] as well as both V2-specific antibodies and T-cell responses in non-human primate studies evaluating poxvirus/gp120 vaccination[55,56]. These responses show both the cellular and antibody response, despite being non-neutralizing, are crucial for protection from SHIV infection.

While needle-free SL/B immunization generated a strong mucosal IgG and IgA vaccine-specific response, these responses contracted significantly at the time of pre-challenge in the rectal

secretions, with most animals showing no detectable vaccine-specific rectal IgA. In contrast, all immunized animals had detectable anti-gp120 IgG in rectal secretions at the pre-challenge time point, and animals showing the greatest level of contraction in the rectal IgG response were significantly more likely to be infected (Supplementary Fig. 5d), highlighting the role of mucosal antibody responses in protection from mucosal pathogens. Thus, it is important to test the influence of different adjuvants on the magnitude and longevity of antibody responses induced by needle-free oral vaccination. The potential adjuvants include TLR7/8 agonists, TLR4 agonists and a combination of these. The development of improved vaccine regimens and adjuvants that improve the duration of the antibody response, especially in the mucosal secretions, may aid in long-term protection from infection. Additionally, unlike the rectal IgA responses, needle-free SL/B immunization resulted in a strong and long lasting HIV-1 specific vaginal IgA response, still detectable at the time of pre-challenge. Previous studies have demonstrated the importance of HIV-1 specific vaginal IgA in protection from vaginal SHIV challenge[5,57], indicating needle-free SL/B may be superior to ID/SC immunization in this context.

In conclusion, our study demonstrates the vaccine-mediated protection of MVA-HIV/cycP-gp120 immunization against a pathogenic, heterologous SHIV, as well as the viability and effectiveness of needle-free SL/B immunization as an alternative to conventional needle-based vaccination. Future studies will further evaluate the needle-free SL/B route of vaccination in comparison with conventional routes, including examining the effects of adjuvant dosage on vaccine responses as well as investigating this route in generating protective immune responses against other mucosal pathogens.

## Methods

**Ethics statement**. All housing and experiments involving rhesus macaques were conducted at the Yerkes National Primate Research Center, and protocols were approved by the Emory University Institutional Animal Care and Use Committee (IACUC) protocol YER-2003491. Experiments were carried out in accordance to USDA regulations and recommendations derived from the Guide for the Care and Use of Laboratory Animals. Rhesus macaques were housed in pairs in standard non-human primate cages and provided with both standard primate feed (Purina monkey chow) fresh fruit, and enrichment daily, as well free access to water. Immunizations, blood draws, and other sample collections were performed under anesthesia with ketamine (5–10 mg kg$^{-1}$) or telazol (3–5 mg kg$^{-1}$) performed by trained research and veterinary staff.

**Animals and immunization**. Indian female rhesus macaques were immunized on weeks 0 and 15 with $1 \times 10^8$ pfu of MVA-62Sm expressing HIV-1 Gag, Pol, and Env (strain ADA gp150)[58], and boosted with 100 µg of recombinant JRFL-hCMP-V1cyc (cycP-gp120)[18] in combination with the mucosal adjuvant double mutated heat-labile enterotoxin (dmLT)[13]. MVA-62Sm was grown in chicken embryo fibroblasts. Animals were separated into three groups based on the route of immunization. Group 1, Topical application: Immunizations were applied topically by pipette to the buccal and sublingual tissue in 100 µl volume, with the animals remaining in upright position for one minute to allow for natural absorption. dmLT dose was 25 µg per tissue site for each time point. The dose of MVA-HIV and cycP-gp120 was split between both tissue sites. Group 2, Needle-free oral injection: Immunizations were delivered via the Syrijet Mark-II Needless Injector (Keystone Industries) as 100 µl injections to the sublingual and buccal tissues. MVA-HIV and cycP-gp120 doses were split between both tissue sites. dmLT dose was 25 µg per tissue site for the first cycP-gp120 boost, and 1 µg per tissue site for the second cycP-gp120 boost. Group 3, Intradermal and subcutaneous injection: MVA-HIV was delivered intradermally and cycP-gp120 was delivered subcutaneously to the right and left thigh, the dose being split between each thigh. Both subcutaneous and intradermal injections were delivered at a volume of 100 µl. dmLT dose for each time point was 1 µg per site. Recombinant cycP-gp120 was produced by transfection of 293-F cells (ThermoFisher) with a plasmid encoding secreted JRFL-cycP-gp120, followed by lectin column purification (GE) and buffer-exchange dialysis with PBS[18]. Trimeric cycP-gp120 was confirmed by Native-PAGE gel analysis.

**Intrarectal SHIV challenge**. Rhesus macaques were challenged intra-rectally with HIV-1 clade-B Env expressing neutralization resistant SHIV-SF162P3 diluted 1:100

in 1 ml sterile RPMI weekly for a maximum of six challenges[39]. Challenge stocks were derived from the expansion of NIH AIDS repository derived SHIV-SF162P3 in rhesus PBMCs. Following sequencing, stocks were titrated for low dose repeated intra-rectal challenges. Plasma samples taken one week post each challenge were analyzed by PCR for detection of viremia (>60 copies/ml). The following primers were used to detect SIV gag in plasma, 5′-GCAGAGGAGGAAATTACCCAG TAC-3′ and 5′-CAATTTTACCCAGGCATTTAATGTT-3. Viral loads were monitored for up to 12 weeks post-infection to determine the peak and set-point viral loads. Viral load analysis performed by the Yerkes CFAR Virology Core.

**Binding antibody detection assays**. Rhesus macaque serum IgG binding to recombinant ADA gp120 (Immune Technology Corp) was measured by ELISA. Microtiter plates were coated with 0.5 µg ml$^{-1}$ of ADA gp120 in PBS at 4 °C overnight followed by 3-fold serial dilutions of rhesus macaque sera before detection with anti-rhesus IgG HRP (Southern Biotech) diluted 1:4000 and the TMB Microwell Peroxidase Substrate Kit (KPL). Additionally, serum IgG binding to dmLT was measured by ELISA using microtiter plates coated with 0.5 µg ml$^{-1}$ dmLT in PBS and detected similarly to anti-gp120 IgG. Concentrations of gp120 or dmLT-specific IgG were quantified using a rhesus IgG standard based on microtiter plates coated with 1 µg ml$^{-1}$ anti-rhesus IgG (Southern Biotech) and standardized using recombinant rhesus IgG (Southern Biotech) serially diluted 3-fold, starting with 100 ng ml$^{-1}$ [14]. Peptides corresponding to the V2-hotspot region of HIV-1 strains JR-FL-E168K (N-RDKVQKEYALFYKLD-C), ADA (N-RDKVKKDYAL-FYRLD-C), SF162P3 (N-GNKMQKEYALFYRLD-C) were synthesized (Genemed Synthesis Inc.), microtiter plates were coated with 1 µg ml$^{-1}$ of peptide at 4 °C overnight, and binding of macaque sera (1:100 dilution) was measured by ELISA (OD 450 reading). For mucosal antibody responses, rectal, vaginal or salivary secretions were eluted from Weck-Cel collection sponges[32]. Concentrations of ng-specific IgG or IgA antibodies to gp120 ADA or gp70-V1V2 Clade B/Case A2 (from Dr. Abraham Pinter, Rutgers Medical School, NJ) were measured by ELISA and divided by the concentration of total ug IgG or IgA to obtain the specific activity[59]. Specific activities were considered significant if they were greater than the mean specific activity plus three standard deviations measured in pre-immunization samples.

**Neutralization assays**. Neutralizing antibody activity was measured in 96-well culture plates by using Tat-regulated luciferase (Luc) reporter gene expression to quantify reductions in virus infection in TZM-bl cells. TZM-bl cells were obtained from the NIH AIDS Research and Reference Reagent Program, as contributed by John Kappes and Xiaoyun Wu. Assays were performed with HIV-1 Env-pseudo-typed viruses as described[60]. Test samples were diluted over a range of 1:20 to 1:43740 in cell culture medium and pre-incubated with virus (~150,000 relative light unit equivalents) for 1 h at 37 °C before addition of cells. Following 48 h incubation, cells were lysed and Luc activity determined using a microtiter plate luminometer and BriteLite Plus Reagent (Perkin Elmer). Neutralization titers are the sample dilution at which relative luminescence units (RLU) were reduced by 50% compared to virus control wells after subtraction of background RLU in cell control wells (ID$_{50}$). Serum samples were heat-inactivated at 56 °C for 1 h prior to assay. Positive values were reported as being at least 3x baseline values standardized against the negative control virus, SVA-MLV.

**Binding antibody multiplex assay (BAMA)**. BAMA assays were performed as follows[33,61]. Serial dilutions (starting at 1:80, six five-fold dilutions) of rabbit or rhesus macaque sera were reacted against beads conjugated to a panel of gp120 (strains 51802, BORI, BJOX002, 254008, CNE20, TT31P.2792, B.6240, A244), uncleaved trimeric gp140 (strains RHPA4259, AE.01.con_env03, 1086.C, C. CH505TF, WITO4160, BF1266, 9004s, SC42261), and gp70-V1V2 scaffold proteins (strains B.CaseA, 7060101641, CM244.ec1, TV1.21, 001428.2.42, CAP210.2.00.E8, C2101.c01, BJOX002000.03.2, BF1266_431a, 96ZM651.02, RHPA4259.7, Ce1086_B2, 62357.14, 700010058, 191084_B7, TT31P.2F10.2792) representing the global panel of HIV-1 Envs[35]. Binding of sera to beads was detected using a secondary biotin-conjugated anti-rhesus IgG and measured via Bio-Plex. Binding is reported as Area Under Curve (AUC) analysis generated from measured MFI's at different serial dilutions. Criteria for positive reactivity was as follows: MFI at 1:80 > 100, MFI at 1:80>Ag-specific cutoff (95th percentile of all pre-bleed for study for each antigen), MFI>3-fold that of matched pre-bleed before and after blank bead subtraction.

**Linear epitope mapping peptide microarray**. Solid phase peptide microarray epitope mapping was performed as follows[37]. Briefly, array slides were prepared by JPT Peptide Technologies GmbH (Germany) by printing a library designed by Dr. B. Korber, Los Alamos National Laboratory, onto Epoxy glass slides (PolyAn GmbH, Germany). The library contains 15-mer peptides overlapping by 12, covering consensus Env (gp160) clade A, B, C, D, Group M, CRF1, and CRF2 and vaccine strains (gp120) 1.A244, 1.TH023, MN, C.1086, C.TV1, and C.ZM651. Sera were diluted 1:50 and applied to the peptide array, followed by washing and detection using goat anti-human IgG-Alexa Fluor 647. Array slides were scanned at a wavelength of 635 nm with an InnoScan 710 AL scanner (Innopsys, France) using XDR mode. Scan images were analyzed using MagPix 8.0 software to obtain

binding intensity values for all peptides. Microarray data were then processing using R package pepStat[36] to obtain binding signal for each peptide, which is defined as $\log_2$ (Intensity of post-immunization sample/intensity of matched baseline sample). Binding magnitude to each identified epitope is defined as the highest binding signal by a single peptide within the epitope region.

**Antibody-dependent cellular viral inhibition (ADCVI).** Assays for ADCVI activity in rhesus macaque serum were performed using HIV SF162 infected CEM-NKr cells and cryopreserved human PBMC derived effector cells[62]. Briefly, on day 1, CCR5+ CEM-NK$^r$ cells (a gift from Dr. Jim Hoxie, University of Pennsylvania) were infected with HIV SF162 at an MOI of 0.01. On day 2, cryopreserved PBMC were thawed, washed, and added to wells of V-bottom plates in 100 µl of medium containing $1 \times 10^5$ cells. On day 3, 50 µl of 1:25 diluted sterile-filtered serum and 50 µl containing $1 \times 10^4$ washed, infected CEM-NK$^r$ cells were added to each well. A pool of monoclonal antibodies was used as a positive control. On day 7, the cells were washed twice to remove any gag-specific antibodies which may interfere with the assay readout On day 10, the medium was harvested, treated with TritonX-100 detergent, and analyzed for viral content using a gag-p24 capture ELISA[63]. The % inhibition of infection was determined by comparing the p24 concentration in wells that had been incubated with test serum to that in wells incubated with naive control macaque serum.

**Antibody-dependent phagocytosis (ADP).** Phagocytosis by serum antibodies was measured using the THP-1 monocytic cell line (ATCC) and fluorescent beads labeled with gp120 SHIV$_{162P3}$ (Immune Technology). The THP-1 cells were maintained at a low density ($<0.5 \times 10^6$ ml$^{-1}$) in RPMI 1640 medium containing HEPES, 100 units ml$^{-1}$ pencillin, 100 µg ml$^{-1}$ streptomycin, 0.3 mg ml$^{-1}$ L-glutamine, 1 mM sodium pyruvate and 10% FBS (all Gibco). For the ADP assay, 20 µl of 1 µm neutravidin-labeled Fluorospheres (Invitrogen) were washed in 1 ml of sterile 1% BSA in PBS in a sterile 1.5 ml microcentrifuge tube that had been preblocked with BSA overnight at 4˚C. The beads were resuspended in 200 µl of 1% BSA containing 7 µg of biotinylated anti-HIS tag mouse monoclonal antibody (ThermoFisher) and mixed at 1100 rpm for 1 h at room temperature. The beads were then washed, resuspended in 200 µl of 1% BSA containing 14 µg of HIS-tagged gp120 SHIV$_{162P3}$ protein (Immune Technology), and mixed at 1100 rpm overnight at 4 ˚C. The following day, beads were washed and resuspended in 5.5 ml medium. The beads were added in a 25 µl volume to wells of V-bottom plates containing 25 µl of diluted sterile-filtered test serum, pooled serum from naive control rhesus macaques or a positive control anti-HIV Env human monoclonal IgG antibody (3BNC117; AIDS Reagent Program). Serum samples were tested in triplicate at 5-fold dilutions starting with a 1:20 final dilution. After a 30 min incubation at 37˚C and 5% CO$_2$, $2 \times 10^4$ THP-1 cells in 50 µl were added to each well. The plates were then incubated for 5 h. The assay was terminated by centrifuging the plates at 300× $g$ for 5 min, removing the medium, and resuspending the cells in 200 µl of 1% paraformaldehyde. After storage at 4 ˚C overnight, the cells were analyzed for fluorescence by flow cytometry. To determine the phagocytic score, the % of fluorescent (bead-positive) cells was first multiplied by their median fluorescence intensity. This value was then divided by the value obtained for naive control serum at the same dilution. Results are presented for 1:20 serum dilutions, which produced peak results. The 3BNC117 antibody (NIH AIDS Reagent Program) produced a score of 6.5 at 1 µg ml$^{-1}$.

**Flow cytometry and T-cell responses.** PBMCs were stimulated with Gag (clade-B consensus) or Env (clade-B consensus) peptides (2 µg ml$^{-1}$ final each) (NIH AIDS Reagent Program) along with anti-CD28 (BD) and anti-CD49d (BD) co-stimulatory molecules (1 µg ml$^{-1}$ final each) for two hours before the addition of GolgiStop (BD) and GolgiPlug (BD) and incubated for an additional four hours. PBMCs were then surface stained with antibodies against CD3 (BD, SP34-2, Cat# 562381, 1:400 dilution), CD4 (BD, SK3, Custom, 1:2000 dilution), CD8 (BD, RPA-T8, Custom, 1:600 dilution), and LIVE/DEAD Fixable stain (ThermoFisher), then permeablized with Cytofix/cytoperm (BD, Cat# 554714) and stained with IFN-γ (BD, B27, Cat# 560371, 1:200 dilution), TNF-α (BD, MAb11, Cat# 561023, 1:200 dilution), and IL-2 (BD, MQ1-17H12, Cat# 560707, 1:200 dilution). Rectal Biopsy samples were digested with 200 U ml$^{-1}$ Collagenase-IV (Worthington) and 0.03% DNAse-I (Life) for two hours before mechanical disruption with a syringe and needle followed by washing with RPMI supplemented with 10% FBS and stained for phenotypic markers, CD3 (BD, SP34-2, Cat# 558124, 1:200), CD4 (BD, SK3, Custom, 1:2000 dilution), CD8 (BD, RPA-T8, Custom, 1:600 dilution), CD45RA (BD, 5H9, Cat# 561216, 1:40 dilution), CCR7 (BD, 150503, Cat# 562381, 1:50 dilution), CCR5 (BD, 3A9, Cat# 742913, 1:40 dilution), HLA-DR (BD, G46-6, Custom, 1:500 dilution), and LIVE/DEAD for flow cytometry analysis.

**Rhesus macaque tissue digestion and flow cytometry.** Uninfected rhesus macaques scheduled for necropsy were euthanized and their buccal tissue, sublingual tissue, and submandibular, submental, and inguinal lymph nodes were collected. Paraffin-embedded tissues were sectioned and stained with Hematoxylin and eosin or characterized via immunohistochemistry with an anti-Langerin monoclonal antibody (Adcam) and visualized with a light microscope. A single-cell suspension was made by digestion of minced tissue with HBSS supplemented with 200 U ml$^{-1}$ Collagenase-IV (Worthington) and 0.03% DNAse-I (Life) for two hours before mechanical disruption with a syringe and needle (gauge 12), passage through a cell strainer, and washed with RPMI + 10% FBS. Alternatively, buccal and sublingual tissues were paraffin-embedded and analyzed via H&E stain or immunohistochemistry with an anti-Langerin monoclonal antibody (Adcam, Cat# ab192027). Single-cell suspensions were analyzed via flow cytometry with the following anti-human monoclonal antibodies: CD209 (BD, DCN46, Cat# 564127), CD1c (Biolegend, L161, Cat# 331515), CD123 (BD, 7G3, Cat# 562391), CD45 (BD, D058-1283, Cat# 563861), CD14 (BD, M5E2, Cat# 564055), CD16 (BD, 3G8, Cat# 563691), CD3 (BD, SP34-2, Cat# 561805), CD20 (BD, 2H7, Cat# 560631), and LIVE/DEAD Fixable stain (ThermoFisher). All antibodies used at a dilution of 1:40. Characterization of retinoic acid production in macaque cells was performed using the ALDEFLUOR kit (StemCell) per the manufacturer's instructions. After incubation with the ALDEFLUOR reagent in test and control settings, cells were stained and analyzed via flow cytometry, ALDEFLUOR$^+$ cell gating was based on the control samples. %ALDEFLUOR$^+$ cells were calculated by subtracting the ALDEFLUOR$^+$ cell frequency in the control sample gate from the test sample gate.

**Statistical analysis.** Statistical analysis was performed using Graphpad Prism v7.0. Additional univariate correlation analysis between number of challenges and individual immune measurements was by Kendall's Correlation test performed using R and subsequently confirmed using SAS. Principal component analysis (PCA) was performed in R using the prcomp package. Variables for PCA were preselected by ranking of separation between groups, which is defined as the between-group variance divided by within-group variance for each variable.

**Reporting Summary.** Further information on experimental design is available in the Nature Research Reporting Summary linked to this Article.

## Data availability
All data associated with this manuscript are available upon request.

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

## Acknowledgements

We would like to thank Stephanie Ehnert, Christopher Souder, Dara Johnston, John Wambua, and other members of Yerkes Research Resources for animal care and sampling; Sherrie Jean and the Yerkes veterinary staff for administrating the vaccinations; Shan Liang and Shelly Wang and the CFAR Immunology Core for plasma viral load assays; Deepa Kodandera, Evan Dessasau III, Sanjeev Gumber, Kay Lee Summerville, and other members of the Yerkes Pathology Core for help with Immunohistochemistry experiments; Kiran Gill, Barbara Cervasi and CFAR Immunology Core for maintenance of flow cytometers. This work was supported in part by National Institutes of Health Grants U19AI109633, R01DE02633 (to R.R.A.), Emory University CFAR grant P30 AI050409 and NCRR/NIH base grants P30 RR00165 (to Y.N.P.R.C.). Support for neutralizing antibodies, binding and peptide microarray provided by NIH/NIAID Contract

No. HHSN27201100016C (to D.C.M., C.L.B., S.X.S., G.D.T.). Partial support for production of MVA was provided from the Division of Intramural Research, NIAID (to B.M., L.S.W.).

## Author contributions

A.T.J., R.V. and R.R.A. designed the studies. A.T.J. led the animal studies, processed samples, performed assays, and interpreted data. X.S. and G.D.T. performed BAMA experiments, linear peptide array mapping, and PCA modeling. K.L.W. and P.A.K. performed mucosal antibody characterization, ADP, and ADCVI assays. C.C.L. and D.C. M. performed neutralizing antibody assays. L.S.W. and B.M. generated and provided the MVA-HIV. D.H.B. provided the SHIV-SF162P3, J.D.C. provided the dmLT adjuvant, R. V. provided plasmid encoding JRFL-cycP-gp120 and contributed to discussions. A.T.J. and R.R.A. led the studies and wrote the paper along with all co-authors.

## Additional information

**Competing interests:** The authors declare no competing interests.

