## [Peer Review File · Nature Communications]

Reviewers' Comments:

Reviewer #1:

Remarks to the Author:

This paper reports interesting and novel data on the immunogenicity and relative efficacy in the SHIV macaque model of a vaccine based on recombinant MVA combined with a novel trimeric gp140 protein, tested earlier on for immunogenicity in macaques and rabbits.

The authors also investigate novel routes of delivery such as Intra-dermal/subcutaneous (ID/SC) and sublingual/ buccal (SL/B) administered either as needle free injector or topically.

This study is not a strict comparison of routes because a 25 fold more dmLT adjuvant was used in the first boost in the SL/B route than in the ID/SC route.

The SL/B route and /or the higher dose of the adjuvant used in the first boost immunization elicited overall increased levels and breadth of the antibody response to the envelope and to V2.

Both the needle free SL/B and ID/SC but not the topical SL/B decreased the risk of SHIV-SF162P3 acquisition.

Correlate analysis revealed that ADCVI, and serum antibodies to the V2 hotspot correlated with the risk of virus acquisition when the two vaccinated needle free SL/B and ID/SC were combined.

Overall although, the study was performed with a small number of animals, and the correlate analysis is somehow preliminary, it provides very interesting clues about possible protective responses. The novel immunization method simplify vaccine administration and induce excellent mucosal responses. The protein used in the boost is novel and promising.

Suggestion for improvements:

1) In Fig 2D, 3A to 3H it would be more informative for the reader to write the immunogen used in the various panels (gp120 ADA, gp70 V1/V2 Case a2 etc...) to allow an immediate understanding

2) Detail on the challenge stock needs to be added. It is only stated that he was obtained from Dan Borouch and used at 1:100 dilution

The TCDI-50, or any other measurement of dose and, very importantly, the method of propagation of this virus (in human or macaque cells) needs to be reported.

3) In case the challenge virus was grown in human cells it needs to be clarified if the MVA was grown in chicken embryo fibroblasts or Vero cells or else.

4) In fig 6F it is unclear whether the two animals uninfected (UI) were included since the X axis differs from Fig 6E and 6G.

5) The author may reference two recent papers directly relevant to the findings presented here that also found T cells and antibodies to V2 as correlate of acquisition in NHP using poxvirus based/env protein vaccines.

(Vaccari et al. Nature Medicine 2016;

Vaccari et al. Nature Medicine 2018)

Reviewer #2:

Remarks to the Author:

This manuscript describes cellular and humoral responses to sublingual and buccal vaccination using a needle free injector to deliver vaccines and adjuvant compared to sublingual (topical) and

intra-dermal/subQ vaccination. Both systemic and mucosal (rectal, vaginal and salivary) antibody levels were assessed. The manuscript also describes levels of DC and lymphocytes in the buccal and sublingual tissues. The study demonstrates remarkable results showing durable and potent antibody responses to the needle free vaccination. Results demonstrate needle free (NF) vaccination induced stronger and more durable antibody responses across the board in systemic and mucosal tissues, especially to the V1/V2 and other regions important for HIV vaccine design. The study also demonstrates equivalent cellular immune responses to this vaccination compared to other routes. After vaccination, animals were challenged with SHIV and intra-dermal and needle free vaccinated animals demonstrated delayed acquisition of virus suggesting partial protection. Neutralizing antibodies as well as ADP and ADCVI levels were also assessed, the latter which correlated with delayed acquisition of virus. The paper clearly demonstrates that needle free vaccinations especially in the oral cavity lead to broad and potent antibody responses comparable to or superior to other methods of vaccination, especially in mucosal tissues. In general, these are all key findings important to HIV vaccine design and delivery. This is also to my knowledge the first report of comparative antibody and cellular responses induced from a needle free mucosal vaccination for HIV.

Minor comments:

Page 3 states GALT is located in the small intestine but more accurately this should state that GALT is in the distal small intestine (ileum) as the jejunum typically lacks GALT.

The volume of each injection (needle free, subQ/dermal etc) as well as the volume of virus challenge used for rectal inoculations should be stated in the methods

Figure 4E. What timepoint was examined for this figure?

ADP levels appear higher in NF compared to SL vaccination and yet fig 6E shows a strong correlation between ADCVI and resistance to challenge. Was there a correlation between protection and ADP levels?

Reviewer #3:

Remarks to the Author:

This manuscript describes the immunogenicity of an HIV-1 vaccine regimen delivered via the oral cavity (buccal and sublingual) as a topical application of the vaccine or delivered with a needle-free injector. The immunogenicity and protective efficacy of the oral vaccine regimens were compared to a control vaccine regimen delivered by traditional parenteral routes (intra-dermal/subcutaneously). The results demonstrate that needle-free sublingual/buccal (SL/B) immunization delivered by injector induced serum anti-gp120 IgG responses comparable to those induced by intra-dermal/subcutaneous immunization. However, topical SL/B application of the vaccine induced minimal anti-gp120 IgG serum responses, as expected. The findings demonstrate the benefit of the needle-free injector since all vaccines used the same immunogen doses (with different adjuvant doses as clearly discussed by the authors).

The results described in this manuscript are novel and impactful since most mucosal vaccine delivery strategies utilize topical application of the vaccine. As demonstrated by the topical SL/B immunization in this manuscript, topical mucosal immunization rarely induces serum IgG responses comparable to those induced by parenteral immunization when using the same antigen dose. Others have evaluated oral/sublingual vaccination for HIV in NHP models but they often utilize topical oral/sublingual immunization combined with a parenteral route of immunization and therefore the results in this manuscript are unique in that SL/B immunization was alone, without combining SL/B immunization

with a parenteral prime or boost.

Specific comments for the authors are listed below.

1. This is a nicely written manuscript. The results are clearly presented. The Discussion is clear and highlights important points for future studies.

2. The authors comment in the Discussion that "...animals showing the greatest level of contraction in the rectal IgG response were significantly more likely to be infected (data not shown), ...". The authors also comment "...most animals showing no detectable vaccine-specific rectal IgA." at the time of challenge. Figure 3A demonstrates that the ID/SC vaccine regimen also induced elevated rectal IgG. How did the antigen-specific rectal IgG responses in the ID/SC group correlate with protection? Since the ID/SC group had delayed acquisition of infection and not all animals became infected while all animals in the needle-free injector SL/B group became infected (with lower rectal IgG than the needle-free SL/B group and no rectal IgA) it would be helpful if the authors could further discuss their thoughts on the role of mucosal IgG and mucosal IgA on vaccine-induced protection against mucosal HIV-1 transmission. Despite the use of an effective method of mucosal immunization, such as the needle-free SL/B injector, if the vaccine-induced IgA responses do not persist and if the parenteral vaccine also induces mucosal IgG, how do mucosally-administered vaccines fit in the plan for an HIV-1 vaccine?

3. Have the authors evaluated the needle-free SL/B injector vaccine in other animal models as a means to evaluate the reproducibility of the results observed in the current manuscript?

We thank the Reviewers for their insightful comments. We are pleased to know that all three Reviewers like our study and findings. We have considered all of their comments/concerns and revised the manuscript accordingly. Below is our point-by-point response (in RED). We hope that the Reviewers are satisfied with the revised manuscript and the revised manuscript will be accepted for publication in Nature Communications.

Response to Reviewer 1:

Suggestion for improvements:

1) In Fig 2D, 3A to 3H it would be more informative for the reader to write the immunogen used in the various panels (gp120 ADA, gp70 V1/V2 Case a2 etc...) to allow an immediate understanding

We thank the Reviewer for this suggestion. We have updated Fig 2D and Fig 3A-H to specify target antigen used in the assay.

2) Detail on the challenge stock needs to be added. It is only stated that he was obtained from Dan Barouch and used at 1:100 dilution. The TCDI-50, or any other measurement of dose and, very importantly, the method of propagation of this virus (in human or macaque cells) needs to be reported.

The method section detailing the SHIV challenge has been updated to include information about the preparation of the SHIV162P3 stock. In the referenced study, the generation of this stock was highlighted, including the titration and sequencing of Env. In our study, the kinetics of infection in the unvaccinated animals is similar to the referenced study in which more than 50% of animals were infected following the first challenge. The updated methods section states the following:

“Rhesus macaques were challenged intra-rectally with HIV-1 clade-B Env expressing neutralization resistant SHIV-SF162P3 diluted 1:100 in 1ml sterile RPMI weekly for a maximum of six challenges as previously described (Barouch et al., 2013). Briefly, SHIV-SF162P3 derived from the NIH AIDS repository was expanded in rhesus PBMCs, sequenced, and titrated for low dose intra-rectal challenges. Plasma samples taken one week post each challenge were analyzed by PCR for detection of viremia (>60 copies/ml). Viral loads were monitored for up to 12 weeks post infection to determine the peak and set-point viral loads.”

3) In case the challenge virus was grown in human cells it needs to be clarified if the MVA was grown in chicken embryo fibroblasts or Vero cells or else.

As indicated in response to Q#2 above, the challenge virus was grown on rhesus PBMC. In this study the MVA was grown in CEF cells. So, this should be a concern. The methods have been updated to indicate this.

4) In fig 6F it is unclear whether the two animals uninfected (UI) were included since the X axis differs from Fig 6E and 6G.

The discrepancy of the X-axis in the original figure has been changed to be the same between Fig. 6E-G. The two uninfected animals are included in this plot. They have very similar values in TNF- α Env-specific cells, 0.037 and 0.038%.

5) The author may reference two recent papers directly relevant to the findings presented here that also found T cells and antibodies to V2 as correlate of acquisition in NHP using poxvirus based/env protein vaccines. (Vaccari et al. Nature Medicine 2016; Vaccari et al. Nature Medicine 2018)

We appreciate the reviewer for this suggestion. We have updated the discussion to comment on these two studies as they underline the importance of V2-antibodies as well as T-cell responses in both SHIV and SIV challenge models.

Response to Reviewer #2:

Minor comments:

Page 3 states GALT is located in the small intestine but more accurately this should state that GALT is in the distal small intestine (ileum) as the jejunum typically lacks GALT.

We have updated the text to clarify the GALT is primarily located in the distal regions of the small intestine.

The volume of each injection (needle free, subQ/dermal etc) as well as the volume of virus challenge used for rectal inoculations should be stated in the methods

The methods section has been updated to specify the volumes of the immunizations (100 ul) as well as the SHIV challenge (1 ml)

Figure 4E. What timepoint was examined for this figure?

The V2 Hotspot-antibody response in Fig. 4E is measured at WK 25, two weeks post the first protein boost. We updated the figure and legend to clarify this timepoint.

ADP levels appear higher in NF compared to SL vaccination and yet fig 6E shows a strong correlation between ADCVI and resistance to challenge. Was there a correlation between protection and ADP levels?

Interestingly, we did not see a correlation between ADP activity and acquisition of infection (as shown on the right), suggesting additional mechanisms involved with ADCVI are important for protection from SHIV infection. We have updated the discussion to comment on the ADP results

Response to Reviewer #3:

2. The authors comment in the Discussion that “.....animals showing the greatest level of contraction in the rectal IgG response were significantly more likely to be infected (data not shown), ...”. The authors also comment “....most animals showing no detectable vaccine-specific rectal IgA.” at the time of challenge. Figure 3A demonstrates that the ID/SC vaccine regimen also induced elevated rectal IgG. How did the antigen-specific rectal IgG responses in the ID/SC group correlate with protection?

Thank you for raising this point. In Sup. Fig 4B we show that rectal IgG responses at the time of pre-challenge in both needle-free SL/B and ID/SC immunized animals show a positive correlation (trend) with delayed acquisition of infection. However, these trends did not reach significance, perhaps due to the small sample size.

Since the ID/SC group had delayed acquisition of infection and not all animals became infected while all animals in the needle-free injector SL/B group became infected (with lower rectal IgG than the needle-free SL/B group and no rectal IgA) it would be helpful if the authors could further discuss their thoughts on the role of mucosal IgG and mucosal IgA on vaccine-induced protection against mucosal HIV-1 transmission.

The role of HIV-1 specific mucosal IgG and IgA in protection of infection in humans is still somewhat unclear, however multiple studies have demonstrated the importance of mucosal IgG and IgA in animal models. In this study, we found a direct association between rectal IgG and protection, and inverse association between rectal IgA and protection (only in the needle-free group where they were measurable) suggesting IgG may be more important than IgA. However, even in the needle-free group, we observed low levels of IgA. In addition, our group size is relatively small. Due to these limitations we can't make any firm conclusions about the role of IgA in protection.

Despite the use of an effective method of mucosal immunization, such as the needle-free SL/B injector, if the vaccine-induced IgA responses do not persist and if the parenteral vaccine also induces mucosal IgG, how do mucosally-administered vaccines fit in the plan for an HIV-1 vaccine?

It is true that IgA responses do not persist in rectal secretions following needle-free SL/B immunization. However, it is important to note that IgA responses in the vaginal tract are long lasting, and are still detectable at the time of pre-challenge (Fig. 3E). These data strongly support the utility of needle-free oral vaccination to induce a strong IgA response at least in the vaginal secretions. However, we think that optimized adjuvants may boost and increase the longevity of the

rectal IgA response. We have updated the discussion to reflect these points (page 21).

With respect to parenteral vaccine also inducing mucosal IgG, yes it is true. However, the mucosal vaccination did induce markedly higher IgG in rectal secretions compared to parenteral vaccination (Fig. 3A). At pre challenge, we did not see a significant difference because of small group sizes. We think other adjuvants such as TLR7/8 agonists may give stronger IgA response and better persistence leading to stronger rectal IgG responses following mucosal vaccination compared to parenteral vaccination.

3. Have the authors evaluated the needle-free SL/B injector vaccine in other animal models as a means to evaluate the reproducibility of the results observed in the current manuscript?

We are currently investigating other animal models to further characterize needle-free SL/B immunization, however we have not performed any studies as of yet. We think these additional studies are outside the scope of this manuscript. One issue is with small animal models having smaller oral cavities, which makes immunizing with a standard needle-free injector difficult.

Reviewers' Comments:

Reviewer #1:

Remarks to the Author:

The authors have addressed my concerns and suggestions